# An Improved Inverse Beamforming Method: Azimuth Resolution Analysis for Weak Target Detection

**DOI:** 10.3390/s18124160

**Published:** 2018-11-27

**Authors:** Peng Li, Xinhua Zhang, Lanrui Li, Wenlong Zhang

**Affiliations:** 1Acoustic Science and Technology Laboratory, Harbin Engineering University, Harbin 150001, China; xinhua_zh@126.com; 2Key Laboratory of Marine Information Acquisition and Security (Harbin Engineering University), Ministry of Industry and Information Technology, Harbin 150001, China; 3College of Underwater Acoustic Engineering, Harbin Engineering University, Harbin 150001, China; 4Department of Underwater Weaponry & Chemical Defense, Dalian Navy Academy, Dalian 116018, China; lilanrui@mail.dlut.edu.cn (L.L.); 15604285615m0@sina.cn (W.Z.)

**Keywords:** DOA estimation, azimuth resolution, Toeplitz average, inverse beamforming

## Abstract

The inverse beamforming (IBF) is a mature method to improve azimuth resolution. However, for weak targets it is not applicable as IBF enhances side lobes. In this paper, an improved IBF algorithm is proposed to raise the azimuth resolution under the premise of ensuring the detection ability for weak targets. Firstly, from the point of phase compensation, we analyze the cause of side lobes when IBF is applied. Then the improved IBF algorithm recorded as GIBF (the improved inverse beamforming) is proposed by changing the Toeplitz average into the phase construction. The theoretical derivation and simulation data processing show the proposed method can improve the resolution of the *N* sensors to the standard of 2*N* − 1 sensors under different signal-to-noise ratios. Compared with IBF, GIBF has great advantages in detecting weak targets. Passive sonar data are used to further verify the advantages of GIBF; the trajectories on azimuth history diagrams become clear, the azimuth resolution is improved, and the detection ability for weak targets is still robust. In addition, GIBF is combined with the common DOA (direction of arrival) estimation algorithms, such as conventional beamforming and minimum variance distortionless signal response, which proves the applicability of the algorithm.

## 1. Introduction

In passive sonar azimuth estimation research, the most robust method is conventional beamforming (CBF). However, its resolution is limited to array aperture. Based on the study of CBF, scholars have proposed many high-resolution algorithms, such as minimum variance distortionless signal response (MVDR) [1] and multiple signal classification algorithm (MUSIC). These well-known adaptive algorithms [2] do not use all the sound field information. In 1970s, Bucker [3] proposed the inverse beamforming (IBF) as a high-resolution algorithm, which uses sound pressure information in all directions of the sensor’s location. In the 1990s, J.H. Wilson [4] obtained the numerical solution of the integral equation of IBF by the Fourier series method (FSM). In addition, in 1991, J.H. Wilson [5] proposed the Fourier integral method (FIM) which reduces the amount of computation greatly. By applying IBF, the main-lobe width of the beam is narrowed. Thus, the azimuth resolution is improved compared with CBF. The disadvantage is that the side lobes become larger. Since then, research on IBF is mostly on how to suppress side lobes based on the improvement of FIM. Reference [6] provides a side lobe suppression technique which reduces side lobes effectively. However, before applying the method, the calculation of the total acoustic field received by the array and the acoustic field corresponding to side lobes are required, which is hard to obtain in the field. Reference [7] improves the ability of side lobe suppression based on a moving-average method. The expense is the physical length of the array.

Inverse beamforming also has a wide range of applications. Reference [8] applies IBF to synthetic aperture sonar. This combination can effectively improve the detection performance of extended towed array technology. Besides that, the detection performance is no longer greatly reduced at a low signal-to-noise ratio (SNR). In Reference [9], IBF is applied to a circular array. The azimuth resolution and noise suppression capability are greatly improved compared with the CBF. Reference [10] and Reference [11] combine IBF with an adaptive algorithm to obtain a better angle measurement performance than traditional adaptive beamforming at a lower SNR.

Although IBF has many advantages in practical application, it has to face several serious problems, such as the splitting of spectral peaks, the mutual interference between targets, and the submergence of weak targets in the strong target side lobes, especially when the strong and weak targets are relatively close in azimuth. Existing methods for mitigating these problems are post-processing, such as the smoothing algorithm used in References [6,11], such as the method of subtracting the sound field corresponding to the sidelobes used in Reference [7]. However, no one has yet suppressed the side lobes in the IBF algorithm before beamforming. In order to solve this problem fundamentally, we analyze the IBF algorithm from the perspective of the phase structure of the signal and find out the cause of the interference. An improved IBF algorithm (recorded as GIBF) is proposed to eliminate the interference terms and preserve the array extension terms by changing the Toeplitz averaging process. This achieves the ability to increase the azimuth resolution and also suppresses the interference introduced by the IBF algorithm.

The rest of this paper is organized as follows. In Section 2, this paper explains the principle of IBF and analyzes how the side lobes are generated from the phase compensation. Then, in Section 3, a new method is proposed and the performance of CBF, IBF, GIBF under the circumstances of coherent signal sources and incoherent signal sources are analyzed. Section 4 shows the simulation and comparison between IBF, CBF, and improved inverse beamforming (GIBF). Finally, in Section 5, the algorithm GIBF is applied on experimental data and the results showed that the detection capability for a weak target is robust and GIBF is beneficial for the improvement of the azimuth resolution. 

## 2. Analysis of IBF

### 2.1. Principle of IBF

Inverse beamforming is a method to estimate the plane-wave density of the acoustic field by using the covariance matrix of the array data. Considering the incident azimuth angle of the signal as θ, elevation angle as ϕ, P(f,θ,ϕ) is signal power spectral density at frequency *f*. For an *N* element linear array, the cross spectrum $G_{ki}(f)$ of the *k*-th sensor and the *i*-th sensor can be defined as:(1)$$G_{ki}(f)=\int_{0}^{\pi }d\theta \int_{-\pi }^{\pi }d\phi \sin\theta P(f,\theta ,\phi )\exp\left\{-j2\pi f[\tau_{k}(\theta ,\phi )-\tau_{i}(\theta ,\phi )]\right\}\;$$ where, $\tau_{k}(\theta ,\phi )-\tau_{i}(\theta ,\phi )$ is the time delay difference between the two sensors. $G_{ki}(f)$ is denoted as C(l,f), l=k−i. By taking the Fourier integral method [12], the solution of IBF can be obtained as follows:(2)$$FIM(\theta ,f)=\frac{1}{2N-1}\sum_{l=-(N-1)}^{N-1}C(l,f)e^{i2\pi fld\cos\theta_{0}/c}\;$$ where, *c* is the speed of sound, *d* is the hydrophone spacing. The performance is evaluated by References [5,12]. The number of array elements is expanded from *N* to 2*N* − 1. The FIM gives a 3-dB array noise gain advantage over CBF and has a beam width that is 2/3 the CBF beam width. Since the sensor number is increased, the azimuth resolution of IBF is higher than CBF.

### 2.2. Phase Compensation Analysis of IBF Theory

For the *N* element uniform linear array (ULA), if the incident azimuth angle of the plane wave is θ, then the output of the *n*th element becomes:(3)Xn(t)=Acos(wt+nφ)=ARe[ejwt·ejnφ]  where, φ is the phase difference between the adjacent sensor signals. The cross correlation can be expressed as:(4)R(i,l)=Xi(t)·X∗l(t)=A2Re[ej(i−l)φ] 

Then, the Toeplitz average is applied to the correlation matrix *R* to obtain C(n).

(5)C(n)=1N−|n|∑i=0N−1−|n|R(i,i+n)=A2Re[e−jnφ],n>0 

(6)C(n)=1N−|n|∑i=|n|N−1R(i,i+n)=A2Re[e−jnφ],n≤0 

Thus, the solution to IBF integral equations can be given by:(7)$$FIM(\theta ,f)=\frac{1}{2M-1}\sum_{n=-(M-1)}^{M-1}C(n,f)e^{i2\pi \;f\;n\;d\cos\theta_{0}/c}\;$$

The above is the traditional IBF process. Our phase analysis starts with Formulas (5) and (6). Toeplitz average is regarded as the process of reconstructing array data according to the acoustic path difference between array elements. The formula (7) is a superposition of reconstructed array data on the same phase to achieve the DOA estimation.

Formulas (5) and (6) shows that the phase item of C(n) is related to *n* and φ. Consider C(n) as reconstructed array data and *n* is the element ordinal number. The phase difference between reconstructed adjacent sensors is equal to the phase difference between the actual array elements. Thus, Formula (7) can be considered as a CBF process. If the original number of elements is *N*, the number of elements after reconstruction becomes 2*N* − 1. The directivity function $D_{la}(\varphi ,t)$ of CBF for 2*N* − 1 linear array [13,14] can be obtained as:(8)$$D_{la}(\varphi ,t)=\frac{\sin((2N-1)\varphi /2)}{\left(2N-1)\sin(\varphi /2\right)}\;$$

For the reconstruction of the *N*-element, the output of the array is C(n). The directivity index can be solved as:(9)$$D_{IBF}(\varphi ,t)=\frac{\sum_{n=1-N}^{N-1}C(n)}{(2N-1)A^{2}}=\frac{\sin((2N-1)\varphi /2)}{\left(2N-1)\sin(\varphi /2\right)}\;$$

Formula (8)–(9) shows that $D_{IBF}$ of *N*-elements is the same as $D_{la}$ of (2*N* – 1)-elements. It also shows that the main-lobe width obtained by the IBF of the *N-*element ULA is consistent with the width obtained by the CBF of the (2*N* – 1)-element ULA.

### 2.3. Cause of Side-Lobe Enhancement

It was found in the study of IBF simulation that the beamforming output of the *N*-element ULA IBF is the same as that of the (2*N* – 1)-element ULA CBF when only one target exists, which is consistent with the theoretical derivation above. But when there are multiple targets, side lobes of IBF become obviously larger. To explain this phenomenon, this paper assumes two targets on different azimuth. The output of the *m*-element array is:(10)Fm(t)=A1cos(w1t1+(m−1)φ1)+A2cos(w2t2+(m−1)φ2)=A1Re[ejw1t1·ej(m−1)φ1]+A2Re[ejw2t2·ej(m−1)φ2] where, $w_{1}$ is the angular frequency of first target signal, $t_{1}$ is propagation time, $\varphi_{1}$ is the phase difference between two adjacent sensors of the first target; $w_{2}$ is the signal angular frequency of the second target, $t_{2}$ is propagation time, $\varphi_{2}$ is the phase difference between two adjacent sensors of the second target. Then the correlation matrix can be expressed as:(11)R(i,l)=Fi(t)·F∗l(t)=Re[A12ej(i−l)φ1+A1A2ej(w1t1−w2t2+iφ1−lφ2)+A1A2ej(w2t2−w1t1+iφ2−lφ1)+A22ej(i−l)φ2] 

After Toeplitz averaging, the matrix R becomes C(n):

When *n* > 0,

(12)C(n)=1N−|n|∑i=0N−1−|n|R(i,i+n)=1N−|n|∑i=0N−1−|n|Re[A12e−jnφ1+A1A2ej(w1t1−w2t2+iφ1−(i+n)φ2)+A1A2ej(w2t2−w1t1+iφ2−(i+n)φ1)+A22e−jnφ2]

When *n* < 0,

(13)C(n)=1N−|n|∑i=|n|N−1R(i,i+n)=1N−|n|∑i=|n|N−1Re[A12e−jnφ1+A1A2ej(w1t1−w2t2+iφ1−(i+n)φ2)+A1A2ej(w2t2−w1t1+iφ2−(i+n)φ1)+A22e−jnφ2]

The Formulas (12) and (13) show that the existence of cross terms between signals introduces a number of phase components besides $\varphi_{1}$ and $\varphi_{2}$. Because C(n) is a combination of R(i,i+n) with different *i*, it is inevitable that some energy will be superimposed on the non-target azimuth due to the existence of cross terms. For example, when *n* changes from 1 to *N*, R(i,i+n) with *i* = *n* in C(n) is selected. Each of the sets of signal components [R(1,1+1),R(2,2+2),…,R(N−1,N−1+N−1),] corresponds to two cross terms $A_{1}A_{2}e^{j(w_{1}t_{1}-w_{2}t_{2}+i\varphi_{1}-(i+n)\varphi_{2})}$ and $A_{1}A_{2}e^{j(w_{2}t_{2}-w_{1}t_{1}+i\varphi_{2}-(i+n)\varphi_{1})}$, wherein the phase vector corresponding to the first cross term $A_{1}A_{2}e^{j(w_{1}t_{1}-w_{2}t_{2}+i\varphi_{1}-(i+n)\varphi_{2})}$ is $\left[\varphi_{1}-2\varphi_{2}，2(\varphi_{1}-2\varphi_{2}),3(\varphi_{1}-2\varphi_{2}),\ldots ,N(\varphi_{1}-2\varphi_{2})\right]$. This phase vector also satisfies the phase superposition principle of beamforming. Therefore, when beamforming is performed, these cross terms cause the false target to appear in the corresponding azimuth of $\varphi_{1}-2\varphi_{2}$. More broadly, as long as *i* takes an integer multiple of *n*, i.e., *i* = *mn*, where *m* is a positive integer. The first cross term $R_{cross1}$ can be expressed as:(14)$$R_{cross1}=A_{1}A_{2}e^{j(w_{1}t_{1}-w_{2}t_{2}+n(m\varphi_{1}-(m+1)\varphi_{2}))}\;$$

According to Formula (14), interference occurs in the direction corresponding to $m\varphi_{1}-(m+1)\varphi_{2}$. The more targets there are and the more cross terms there are, the more interference will appear on the beam pattern. 

## 3. DOA Estimation Principle of GIBF

### 3.1. GIBF Principle

In order to eliminate unnecessary phase components and reduce sidelobes, this paper proposes an improved IBF method, denoted as GIBF. Because sidelobes come from the term $R_{cross1}$, we improve the IBF from this one. In order to make the signal component only contain the phases $\varphi_{1}$ and $\varphi_{2}$ corresponding to the two targets azimuth, we let *i* = 0. Thus, the cross term $R_{cross1}$ becomes:(15)$$R_{cross1}=A_{1}A_{2}e^{j(w_{1}t_{1}-w_{2}t_{2}-n\varphi_{2})}\;$$

It was found from the phase of Equation (15) that only phase $\varphi_{1}$ remains at this time. In the same way, only phase $\varphi_{2}$ is left in $A_{1}A_{2}e^{j(w_{2}t_{2}-w_{1}t_{1}+i\varphi_{2}-(i+n)\varphi_{1})}$. At this time, only R(i,i+n) of *i =* 0 in C(n) is retained. Further, since –*N* < *i* + *n* < *N*, the number of terms of R(i,i+n) corresponding to each *i* value is 2*N* – 1 − *i*. Therefore, when *i* = 0, the obtained array aperture extensions are the most. Equivalent to the number of array element number changed from *N* to 2*N* − 1, so the azimuth resolution is also improved. We record the improved C(n) is G(n).

(16)G(n)=R(0,n)=Re[A12e−jnφ1+A1A2ej(w1t1−w2t2−nφ2)+A1A2ej(w2t2−w1t1−nφ1)+A22e−jnφ2]，n>0 

(17)G(n)=R(−n,0)=Re[A12e−jnφ1+A1A2ej(w1t1−w2t2−nφ1)+A1A2ej(w2t2−w1t1−nφ2)+A22e−jnφ2],n≤0 

After that, the DOA estimation can be given by:(18)$$Beam(\theta )=\frac{1}{2N-1}\sum_{n=-(N-1)}^{N-1}G(n)e^{i2\pi \;f\;n\;d\cos\theta /c}\;$$

Also, we use the assumption in Equation (10), and two situations are discussed here: (1) signals of two targets which are coherent; and (2) signals of two targets which are incoherent. In the first situation, if we define $w_{1}t_{1}-w_{2}t_{2}=\varphi_{s}$, $\varphi_{s}$ should be a constant. Then, Formulas (16) and (17) can be written as:(19)G(n)=Re[(A12+A1A2ej(w2t2−w1t1))e−jnφ1+(A1A2ej(w1t1−w2t2)+A22)e−jnφ2],n>0 
(20)G(n)=Re[(A12+A1A2ej(w1t1−w2t2))e−jnφ1+(A1A2ej(w2t2−w1t1)+A22)e−jnφ2],n≤0 

For simplification, consider $A_{1}{}^{2}+A_{1}A_{2}e^{j(w_{2}t_{2}-w_{1}t_{1})}$ as one item with the amplitude of $A_{a}$ and the phase of $\varphi_{a}$, $A_{1}A_{2}e^{j(w_{1}t_{1}-w_{2}t_{2})}+A_{2}{}^{2}$ as the item with the amplitude of $A_{b}$ and the phase of $\varphi_{b}$, $A_{1}A_{2}e^{j(w_{1}t_{1}-w_{2}t_{2})}+A_{2}{}^{2}$ as the item with the amplitude of $A_{c}$ and the phase of $\varphi_{c}$ and $A_{1}A_{2}e^{j(w_{2}t_{2}-w_{1}t_{1})}+A_{2}{}^{2}$ as the item with the amplitude of $A_{d}$ and the phase of $\varphi_{d}$. Formulas (19) and (20) can be rewritten as:(21)G(n)=Re[(Aaej(φa−nφ1)+Abej(φb−nφ2)],n>0 
(22)G(n)=Re[(Acej(φc−nφ1)+Adej(φd−nφ2)],n≤0 

As can be seen from the Formulas (19) and (20), the output signals of the reconstructed *N*-1-element array when n>0 and the reconstructed *N*-element array when n≤0 are both corresponding to the target azimuth. When the two sets of array data are put together for CBF processing, the components of each target signal in the two set of arrays cannot be superimposed on the same phase because $\varphi_{a}\neq \varphi_{c}$, $\varphi_{b}\neq \varphi_{d}$. Therefore, the method does not extend the number of elements from *N* to 2*N* − 1 but is equivalent to superimposing the beamforming outputs of two *N* − 1 elements arrays. The beam pattern may be smoother, and the main lobe will be narrower. Because the effective array length was not improved, the azimuth resolution was not improved.

When dealing with two incoherent sources, the cross term in Formulas (16) and (17) is expected to be zero after accumulation of time.

(23)$$\underset{T\to \infty }{\lim}\int_{0}^{T}A_{1}A_{2}e^{j(w_{1}(t_{1}+t)-w_{2}(t_{2}+t)-n\varphi_{2})}+A_{1}A_{2}e^{j(w_{2}(t_{2}+t)-w_{1}(t_{1}+t)-n\varphi_{1})}dt=0\;$$ where, *T* is the signal processing time. Therefore, G(n) can be simplified as:(24)G(n)=Re[A12e−jnφ1+A22e−jnφ2],n=−N+1,−N+2⋯0⋯N−1 

Formula (24) tells us that the signals of all the virtual array elements can be in-phase superimposed. The number of effective array elements is expanded from *N* to 2*N* − 1 so that the azimuth resolution is improved.

### 3.2. Extension of GIBF Method

The GIBF is equivalent to the expansion of the array elements under the condition of the incoherent signal sources. The reconstructed phase information of the signal is consistent with the phase information of the actual signal. Therefore, after the generation of the new array output by GIBF, a high-resolution algorithm, such as MVDR, can still be used for subsequent processing. We can even apply the GIBF algorithm to each output signal obtained by GIBF. The following is a theoretical analysis of the second order GIBF: Assuming $\varphi_{a}=n\varphi_{1}$, $\varphi_{b}=n\varphi_{2}$, Formula (24) can be rewritten as:(25)G(n)=Re[A12ej(−φa−nφ1)+A22ej(−φb−nφ2)],n=0,1,2⋯2N−1 

Regarding G(n) as the new array element output signal, the corresponding $R(0,n_{2})$ and $R(-n_{2},0)$ can be obtained by using the method above, where $n_{2}=-2N+1,-2N+2,\cdots 2N-1$. As shown in the derivation above, the output $G_{G}$ of the array after two GIBF processing is:(26)GG(n2)=Re[(A12+A1A2ej(φa−φb))e−jn2φ1+(A1A2ej(φb−φa)+A22)e−jn2φ2],n2>0 
(27)GG(n2)=Re[(A12+A1A2ej(φb−φa))e−jn2φ1+(A1A2ej(φa−φb)+A22)e−jn2φ2],n2≤0 

In this case $\varphi_{a}$ and $\varphi_{b}$ are basically independent of time. Unlike incoherent signal processing, time integration cannot remove the phase difference introduced by $A_{1}A_{2}e^{j(\varphi_{b}-\varphi_{a})}$. Therefore, high-order GIBF can increases the array gain and narrow the main lobe width while it cannot improve the azimuth resolution again.

## 4. Simulation

### 4.1. Weak Target Detection Capability of GIBF

By the theoretical derivation from Section 2, we proved that if there is only one target, the IBF algorithm has high-target detection performance, such as high array gain and narrow width of the main lobe. When it comes to multiple targets, problems arise, such as side-lobe increasing and cross-term interference. The following simulations in Figure 1 are used to illustrate this point.

In Figure 1a, the simulation conditions are as follows: the bearing angle was 25°, the array element number was 48, Gaussian white noise was added and the SNR was 0 dB. From the simulation results, it was found that the IBF beam main lobe was narrower and the array gain was higher than the CBF beam output. However, in Figure 1b, when the simulation conditions were changed from a single target to three equal-intensity incoherent targets, whose incident azimuth angles were −30°, 25°, and 50°, respectively, although the width of the main lobe was still very narrow, the interference between 25° and 45° and the background noise level between −90° and −45° were significantly higher than those at the CBF treatment. Therefore, if there was a weak target, it may be submerged in the interference caused by the strong target. 

As shown in the simulation results in Figure 2a, the simulated signal amplitude at 25° was 10 times the target amplitude of −30° and 50°. From the IBF beam output, it was difficult to observe two weak targets, which can still be observed on the CBF. This is clearly disadvantageous in practical passive sonar detection applications. Under the same simulation conditions, Figure 2b shows the comparison between GIBF and CBF. It can be found that the GIBF retains the ability to detect weak targets while narrowing the main lobe. This is because the proposed GIBF algorithm does not introduce signal components corresponding to other directions of arrival when doing the array extension based on the array manifold.

### 4.2. Azimuth Resolution of GIBF

The theoretical derivation in Section 3 shows that applying the GIBF algorithm on the *N* element ULA can obtain the azimuth resolution of the (2*N* − 1)-element ULA. The following simulations were used to illustrate the improvement effect of the GIBF algorithm on the azimuth resolution. Firstly, the CBF algorithm of the *N*-element ULA was used as a reference. When the noise was not added, the beamforming output of the (2*N* – 1)-element ULA CBF and the *N*-element ULA GIBF algorithm were compared. The incident angles of the two targets were 0° and 10°, respectively, and *N* was equal to 24. From the comparison of Figure 3a,b, the beam pattern of the GIBF algorithm and the beam pattern of the CBF algorithm with double array aperture stayed the same under ideal conditions. This phenomenon was consistent with our theoretical derivation.

Next, we add white noise to the simulation and select two signals that are close in azimuth. The incident angles are 9° and 10°, and the SNR are −3 dB and −10 dB, respectively (Figure 4). The 48-element ULA and the 95-element ULA are used to test the detection ability of GIBF. 

From the comparison between the CBF and GIBF of the 48-element ULA, the CBF could not separate the two targets of 9° and 10° under the two signal-to-noise ratios. With GIBF, you can clearly separate two adjacent targets. Comparing the GIBF processing results of the 48-element ULA with the CBF of the 95-element ULA, the CBF processing effect of the double-aperture was better than GIBF after adding noise. In particular, the lower the signal-to-noise ratio, the greater the difference on array gain. This is because the GIBF uses the first array element to virtualize the signal, which introduces the correlated noise between the array elements. Therefore, the array gain will be affected.

After verifying the resolution enhancement effect of GIBF, in order to prove the advantage of GIBF compared with IBF in detecting adjacent weak targets. The amplitude of the simulated 9° target signal was five times the amplitude of the 10° target signal. The CBF, IBF, and GIBF algorithms under the SNR of −3 dB and −20 dB were tested on the 48-element ULA, respectively. The simulation results are shown in Figure 5.

It can be seen from Figure 5 that the IBF and the GIBF obtained a narrower main lobe width than the CBF. The advantage of IBF is that the array gain was the highest. However, there was only one peak on the beamforming output, so it was impossible to distinguish between a 9° target and a 10° target. In the beamforming output of GIBF, it was obvious that there was a clear target in the 10° azimuth. Moreover, regardless of the SNR = −3 dB or SNR = −20 dB, the detection performance of the GIBF for 10° weak targets was robust.

### 4.3. Second Order GIBF 

As mentioned above, in fact, GIBF was more like a process of constructing array signals. Therefore, the signal processed by GIBF can be combined with other signal processing algorithms, such as GIBF and MVDR combined in the following experimental data processing. In addition, GIBF can be used again for signals obtained by GIBF processing to obtain higher detection performance, as shown by the simulation results in Figure 6. The simulation conditions are as follows: the two target signals are in the wave direction of 0° and 20°. The azimuth estimation results of the GIBF and the second order GIBF applied on the 24-element line array at −3 dB and −20 dB were simulated. It can be seen from the processing results that when the GIBF was used again, the main lobe width was narrower and the array gain was higher. Especially when the SNR was reduced to −20 dB, the algorithm of the second order GIBF as obviously better on the target detection performance. However, the disadvantage was that using second order GIBF increased the amount of computation.

## 5. Experimental Data Verification

The following are the South China Sea experimental data results processed by CBF, IBF, and GIBF, respectively. The number of sensors was 64 and the processing frequency band was 150–200 Hz. 

According to the theoretical analysis, the IBF algorithm was based on the Toeplitz average of the signal covariance matrix. The covariance matrix itself was the quadratic form of the signal, which makes strong targets stronger and weak targets weaker. For example, on the azimuth history diagram of the IBF method, the trajectory of the strong targets became more precise while the weak targets could hardly be observed. In addition, the IBF method introduced many other phase components, which can be seen from Figure 7d. In the position where there should be no signal, a lot of peaks whose energy was even stronger than many weak targets showed up. Therefore, the detection capability for weak targets was greatly affected. The GIBF method does not have this problem because the method only uses the first element output multiplying the original array output to construct the phase. The amplitude effect of first element output can be eliminated when the reconstructed data is normalized. Thus, the array signal reconstruction by GIBF is not the quadratic form. Moreover, the GIBF method would not introduce phase components in other directions, so the robustness of detection for weak targets does not decrease compared with that of CBF.

In Figure 8, the targets cannot be distinguished by CBF in 590 s, while GIBF can separate them clearly. For example, from studying target trajectory on a azimuth history diagram we can know that, at azimuth −20°, there should be two targets. These two targets gradually separate from the same orientation over time. At 590 s, the two targets cannot be distinguished from the beam pattern of CBF while the two targets can be distinguished from the beam pattern of GIBF. In addition, from the comparison of azimuth history diagram, the target trajectory processed by GIBF is clearer and more accurate. Therefore, compared with CBF, GIBF improved azimuth resolution without losing detecting ability for weak target. 

In the above we have analyzed that the virtual array output obtained by GIBF which can also be used in conjunction with MVDR theoretically. The following results are processed by MVDR and MVDR in conjunction with GIBF (Figure 9):

Azimuth history diagrams of the two methods show that the result of the combination of MVDR and GIBF is clearer and the trajectory is more complete than that of MVDR. In 590th s, MVDR cannot separate targets near −20° and −10° while the combination of MVDR and GIBF can. The experimental results prove that GIBF can be used with MVDR to improve the azimuth resolution.

## 6. Conclusions

In this paper, for inverse beamforming algorithm, the principle of improving the azimuth resolution and the cause of the side lobes increasing are analyzed. In order to improve the azimuth resolution without increasing the side lobes, we proposed a new array data reconstruction algorithm by improving the Toeplitz averaging process by replacing C(n) with G(n) which would not add phase components other than the target path difference to the array output. In terms of contribution to resolution, the algorithm is equivalent to expanding the array aperture by a factor of two. Therefore, the algorithm can improve the azimuth resolution under the premise of ensuring the weak target detection capability. The advantages are verified through theoretical derivation, simulation experiments and experimental data. Moreover, the algorithm is not only limited to CBF, but also can be combined with high-order extension and the MVDR algorithm to achieve the purpose of improving resolution. Finally, the performance of DOA estimation is improved. However, it can be found from the simulation data and experimental data processing results, GIBF loses some array gain compared with IBF. Therefore, how to effectively improve the array gain and obtain higher resolution is the focus of future research.

## Figures and Tables

**Figure 1 sensors-18-04160-f001:**
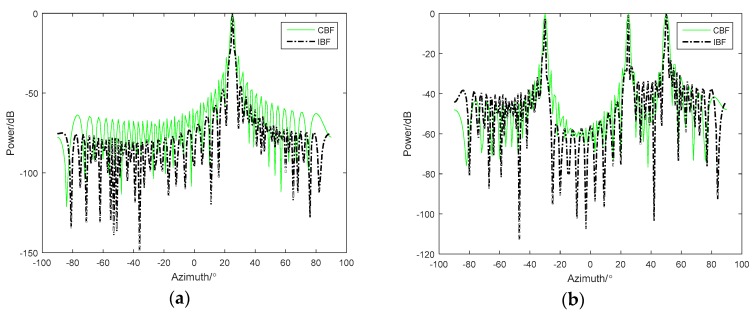
Comparison between conventional beamforming (CBF) and inverse beamforming (IBF) with (**a**) one target and (**b**) three equal energy targets.

**Figure 2 sensors-18-04160-f002:**
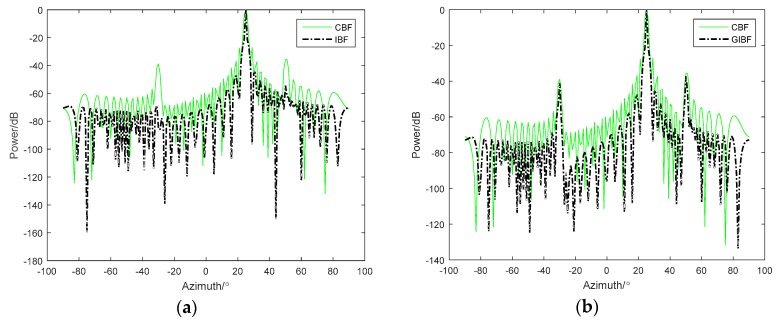
Comparison for three unequal energy targets between (**a**) CBF and IBF and (**b**) CBF and GIBF(the improved inverse beamforming).

**Figure 3 sensors-18-04160-f003:**
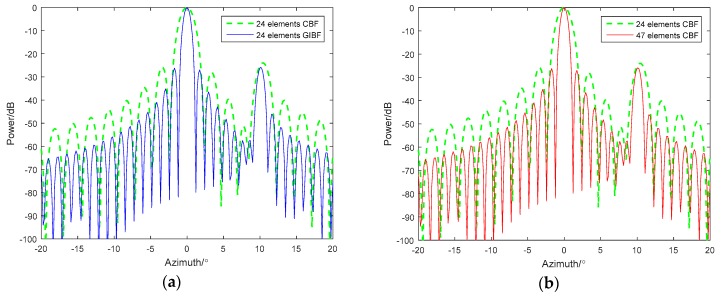
Comparison between (**a**) CBF and GIBF with 24 sensors and (**b**) CBF with a 24-element array and CBF with 47 elements.

**Figure 4 sensors-18-04160-f004:**
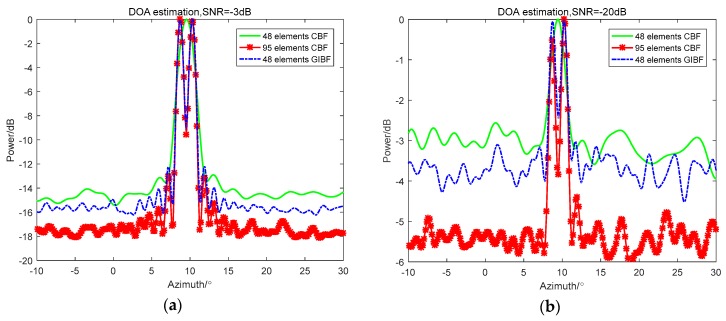
Comparison between CBF and GIBF under different SNRs (signal-to-noise ratio): (**a**) −3 dB; (**b**) −20 dB.

**Figure 5 sensors-18-04160-f005:**
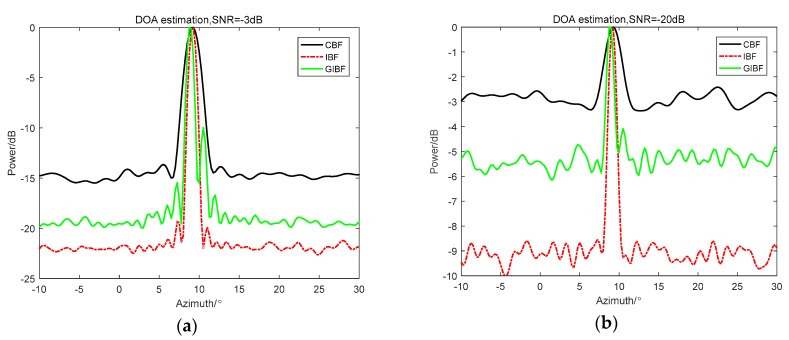
Comparison between CBF, IBF, and GIBF under different SNRs: (**a**) −3 dB; (**b**) −20 dB.

**Figure 6 sensors-18-04160-f006:**
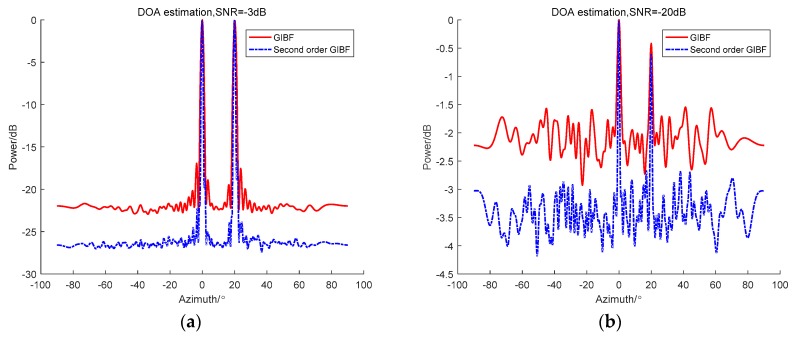
Comparison between GIBF and second order GIBF under different SNRs: (**a**) −3 dB (**b**) −20 dB.

**Figure 7 sensors-18-04160-f007:**
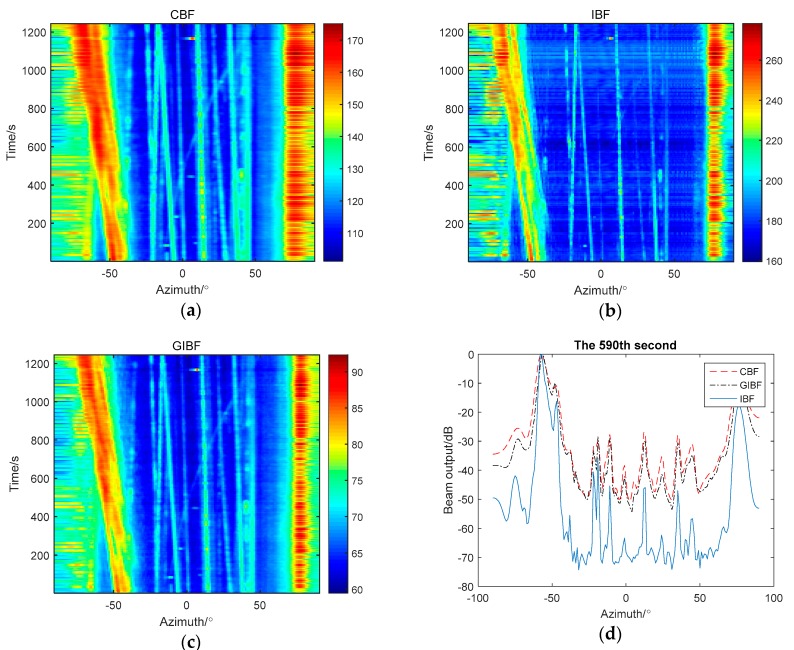
The comparison between three methods: (**a**) azimuth history diagram by CBF; (**b**) azimuth history diagram by IBF; (**c**) azimuth history diagram by GIBF; (**d**) beam pattern of three methods in 590th s.

**Figure 8 sensors-18-04160-f008:**
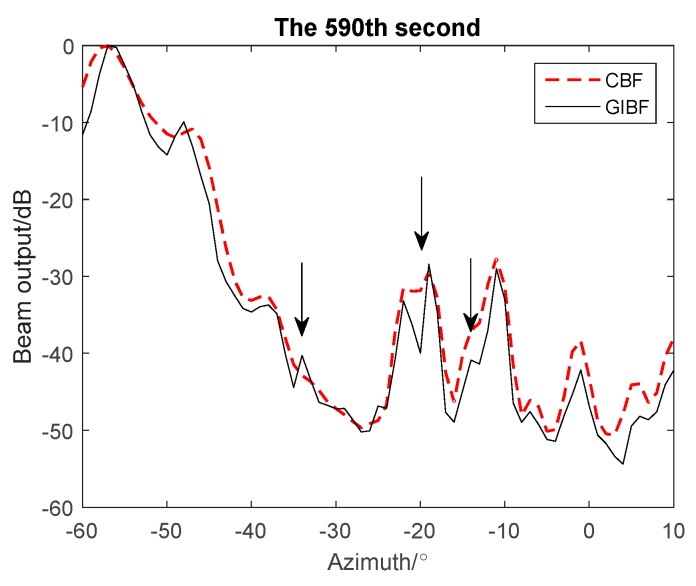
Results of GIBF and CBF processing in 590th seconds of experimental data.

**Figure 9 sensors-18-04160-f009:**
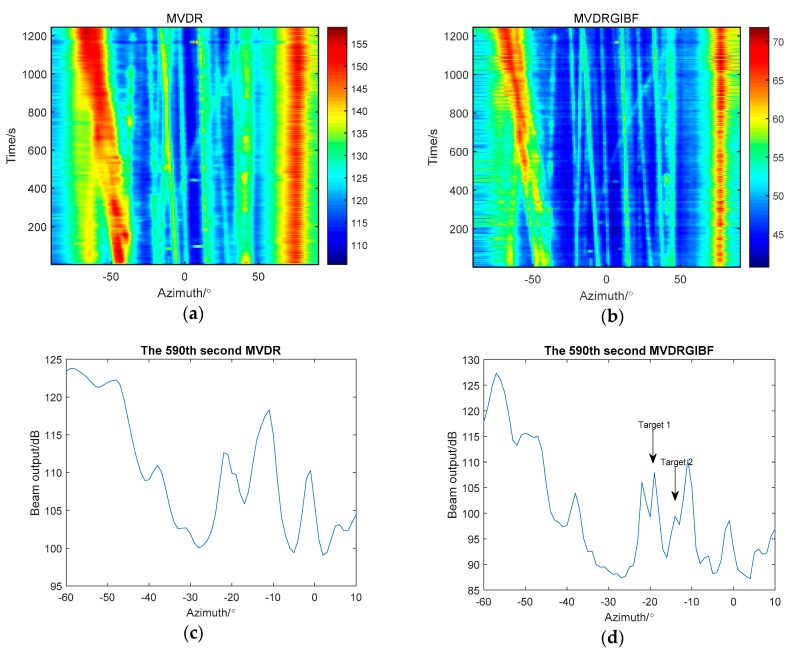
Comparison between applying MVDR (minimum variance distortionless signal response) and combination of MVDR and GIBF: (**a**) azimuth history diagram processed by MVDR; (**b**) azimuth history diagram processed by MVDR and GIBF (**c**) the 590th s results processed by MVDR; (**d**) the 590th s results processed MVDR and GIBF.

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
