# Peer review of "An Improved Inverse Beamforming Method: Azimuth Resolution Analysis for Weak Target Detection"

_sensors, 2018, doi:10.3390/s18124160_

Reviewer 1 Report

In equation 7 please put separation between f, n, d variables.

In line 123, explain more clear GIBF Principle. Why is not neccesary make Toeplitz averaging?

In line 175-175 use the same notation for (a) and (b) graphs.

In line 246, change the first G(n) by C(n).

Author Response

Point 1: For errors that appear in the article

Response 1: Thank you for your guidance and corrections.

I have put separation between f, n, d variables in equation 7 and modified the error in line 175 and line 246.

Point 2: In line 123, explain more clear GIBF principle. Why  is not necessary  make Toeplitz averaging?

Response 2: From the perspective of phase analysis of IBF, the improvement in resolution is mainly due to the Toeplitz average. However, Toeplitz average is also the cause of side lobes. The GIBF algorithm proposed in this paper is to change the Toeplitz averaging process by eliminating the cross terms in which the side lobes are generated and retaining the array extensions that improve the resolution. Thereby achieving the purpose of improving the resolution of weak targets. The Toeplitz averaging process is necessary for IBF, but this paper finds a resolution enhancement algorithm GIBF that is more suitable for weak targets instead of Toeplitz averaging.

In order to better explain this problem, this paper adds more analysis of the causes of side lobes and explains more about GIBF principle in the revision.

Reviewer 2 Report

See attached PDF.

Author Response

Synopsis

I am very grateful for your guidance and advice. Especially the advice on the topic, it is very suitable.

In terms of the content of the article, I added more explanations of the GIBF principle and emphasized that the GIBF was first proposed in this article. The Section IV simulation part was reorganized. Different algorithm simulation experiments were applied under different SNR. The words of the article and figure captions are re-examined.

Please see the attached PDF for details.
